# The Lack of STING Impairs the MHC-I Dependent Antigen Presentation and JAK/STAT Signaling in Murine Macrophages

**DOI:** 10.3390/ijms232214232

**Published:** 2022-11-17

**Authors:** Carmen Caiazza, Teresa Brusco, Federica D’Alessio, Massimo D’Agostino, Angelica Avagliano, Alessandro Arcucci, Concetta Ambrosino, Giuseppe Fiume, Massimo Mallardo

**Affiliations:** 1Department of Molecular Medicine and Medical Biotechnology, University of Naples “Federico II”, Via S. Pansini 5, 80131 Naples, Italy; 2Department of Public Health, University of Naples “Federico II”, Via S. Pansini 5, 80131 Naples, Italy; 3Department of Science and Technology, University of Sannio, Via De Sanctis, 82100 Benevento, Italy; 4IRGS, Biogem-Scarl, Via Camporeale, Ariano Irpino, 83031 Avellino, Italy; 5IEOS-CNR, Via Pansini 6, 80131 Naples, Italy; 6Department of Experimental and Clinical Medicine, University of Catanzaro “Magna Graecia”, 88100 Catanzaro, Italy

**Keywords:** cell biology, STING, JAK/STAT, antigen presentation

## Abstract

STING is a transmembrane ER resident protein that was initially described as a regulator of innate immune response triggered by viral DNA and later found to be involved in a broader range of immune processes. Here, we assessed its role in the antigen presentation by generating a STING KO macrophage cell line. In the absence of STING, we observed an impaired OVA-derived SIINFEKL peptide presentation together with a decreased level of MHC-I complex on the plasma membrane, likely due to a decreased mRNA expression of β2 m light chain as no relevant alterations of the peptide-loading complex (TAPs) were found. Moreover, JAK-STAT signaling resulted in impaired STING KO cells following OVA and LPS treatments, suggesting a dampened activation of immune response. Our data revealed a new molecular role of STING in immune mechanisms that could elucidate its role in the pathogenesis of autoimmune disorders and cancer.

## 1. Introduction

STimulator of INterferon Genes (STING) is a transmembrane ER resident protein involved in the interferon response to infection by pathogens, including bacteria and viruses. STING is a protein of 42 kDa that localizes in the endoplasmic reticulum and is composed of a short N-terminal cytosolic domain, four transmembrane domains, a cytosolic cyclic di-nucleotide (CDN) domain, and a C-terminal tail (CTT) [1]. In resting state, STING is retained in the ER through its interaction with the Ca^2+^ sensor stromal interaction molecule 1 (STIM) [2]. In the cGAS-STING canonical pathway, pathogen infection leads to the activation of cytosolic DNA sensor cGAS (cyclic GMP-AMP (cGAMP) synthase), which synthesizes cyclic guanosine monophosphate–adenosine monophosphate (cGAMP). cGAMP acts on STING, promoting its binding to TANK-binding kinase 1 (TBK1), which next activates the transcription factor IRF3 [3]. IRF3 dimerizes and enters the nucleus to initiate a type I Interferon (IFN-I) response, leading to the expression of a set of interferon-stimulated genes (ISGs) and the establishment of antimicrobial immunity [4]. In addition to IRF3, the activation of cGAS-STING signaling also leads to NF-κB activation, whose regulation relies upon TNF receptor-associated factor 6 (TRAF6), NF-κB essential modulator (NEMO), IKKβ, and TBK1 [5,6].

Major Histocompatibility Complex (MHC) mediates the connection between innate and adaptive immunity. The loading of small peptides (antigens) onto the MHC complex is a fundamental prerequisite to generate a specific response against pathogens [7]. Depending on the origin of the antigen, adaptive immunity can be triggered by two types of MHC complexes. MHC class I (MHC-I) complexes are expressed by all the cells and represent a “kill me” signal that activates the T-CD8+ cytotoxic response [8]. Antigens loaded onto MHC-I originate from cytoplasm and derive from pathogens that have infected the cell or constitute neoantigens produced by tumors. Only professional antigen-presenting cells (APCs) (including dendritic cells, phagocytes, and B lymphocytes) express MHC class II (MHC-II) complexes, which activate the CD4+ T-cell response. Upon processing via endosomal signaling, some antigens may escape the endosome, enter the ER, and associate with MHC-I in a process named cross priming [7].

The MHC-I complex is composed of a heavy chain (H2-K, -L, and -D in mouse and HLA-A, -B, and -C in human) and a β2-microglobulin (β2 m) light chain [9]. Proteasome-dependent proteolysis generates short peptides (8–10 amino acids) that are transported into the ER and loaded onto MHC-I. This process is regulated by a multi-protein complex named the peptide-loading complex (PLC) [9]. The transport into the ER is mediated by the Transporter associated with Antigen Processing (TAP) proteins. TAP is a heterodimeric complex formed by TAP1 and TAP2 [10]. Tapasin (TAPN) functions as a link between the TAP complex and MHC-I chain. In particular, the TAPN complex maintains the MHC-I complex on the ER membrane, allowing the loading of the antigen [11]. Other proteins, including Calnexin (CLX), Calreticulin (CLR), and protein disulfide-isomerase A3 (ERp57), act as chaperons in mediating the quality control of the process [12]. If the peptide entering the ER is longer than 10 amino acids, the ER aminopeptidases ERAP1 and ERAP2 generate the antigen of the correct size [13].

In the present work, we raised the hypothesis of an upstream effect exerted by STING in the MHC-I-dependent antigen presentation. To this end, we analyzed the effect of the depletion of STING in the murine macrophage cell line J774 on the presentation of SIINFEKL peptide on MHC-I after OVA treatment. We found a reduction of OVA-derived SIINFEKL peptides loaded onto MHC-I as well as a decreased level of MHC-I complexes on the plasma membrane in STING KO compared to wild type cells. Moreover. JAK-STAT signaling resulted in impaired STING KO cells following LPS and OVA treatments, suggesting a dampened activation of immune response. Our data revealed a new molecular role of STING in immune mechanisms that could elucidate its role in the pathogenesis of autoimmune disorders and cancer.

## 2. Results

### 2.1. The Lack of STING Impairs the MHC-I-Dependent Antigen Presentation

We asked whether STING could affect the MHC-I-dependent antigen presentation. To this end, we generated STING KO and CTRL J774 murine macrophage cell lines by using the CRISPR/Cas9 system expressing two RNA guides directed against the STING gene (named RNA guide #1 and RNA guide #2) or a scramble sequence, respectively. Next, we performed the chicken ovalbumin (OVA) internalization, processing, and presentation assays. In more detail, by mimicking an infection from an exogenous host, OVA is internalized into the cells and processed in a pathogen-like response, producing a specific epitope (SIINFEKL) loaded onto MHC-I on the plasma membrane [14,15,16,17,18]. We treated J774 CTRL and J774 STING KO cells lines with OVA or PBS as a negative control for 24 h. Next, we measured the amount of the MHC-I-SIINFEKL complex by cytofluorimetric analysis using a PE-conjugated anti-MHC-I-SIINFEKL antibody. We observed that the amount of MHC-I-SIINFEKL complex on the membrane of J774 STING KO cells was strongly reduced compared to that of J774 control cells (Figure 1A–C). No differences were found by using an Ig isotype antibody as a control for non-specific binding (Figure 1A, lower panel). As a further control, we analyzed the efficiency of single RNA guides in silencing STING expression and in reducing the amount of MHC-I and MHC-I-SIINFEKL complexes upon OVA treatment. Thus, we generated two additional STING KO cell lines by STING-specific RNA guide #1 and RNA guide #2. In all the STING KO cell lines (expressing RNA guide #1, RNA guide #2, or a combination of RNA guide #1 and RNA guide #2), the expression of STING was completely and similarly abolished (Appendix A); in addition, upon OVA treatment, the amount of MHC-I-SIINFEKL on the plasma membrane was dramatically reduced (Appendix A) as well as the amount of MHC-I (Appendix A).

### 2.2. The Lack of STING Does Not Affect the Entry and Proteolysis of OVA Exogenous Protein and the Peptide-Loading Complex

Following the observation of the reduced amount of MHC-I-SIINFEKL complex on the membrane of J774 STING KO (Figure 1A–C), we asked whether a defective uptake and proteolysis of OVA could occur in J774 STING KO cells. To this end, we evaluated the uptake of Alexa 488-conjugated ovalbumin after incubation for 30 min at 37 °C or 4 °C as a negative control (Figure 2A,B) by measuring the intensity of fluorescence with a cytofluorimeter. Interestingly, we did not observe any statistically significant differences in the internalization of Alexa 488-conjugated ovalbumin in STING KO and CTRL J774 cells (Figure 2A,B), even though there appeared to be a slight decrease in the internalization of OVA, which could likely suggest a slight impairment of endocytosis due to the lack of STING (Figure 2B). Next, we analyzed the proteolysis of DQ-OVA in STING KO and CTRL J774 after a pulse with 50 μg/mL of DQ-ovalbumin for 3 min and a chase for 30 min at 37 °C or 4 °C as a negative control. DQ-OVA is a self-quenched conjugate of ovalbumin that exhibits bright green fluorescence upon proteolytic degradation and has been specifically designed for the study of antigen processing and presentation [19]. We did not observe any statistically significant differences in the processing of DQ-OVA between CTRL and STING KO J774 cells (Figure 2C,D). Taken together, these results indicate that the reduction of an amount of MHC-I-SIINFEKL complex on the membrane of J774 STING KO cells does not rely upon an impaired uptake and proteolysis of DQ-OVA (Figure 2A–D).

Therefore, we asked whether the absence of STING could affect the peptide loading onto MHC-I. To this end, we analyzed the intracellular formation of MHC-I-SIINFEKL complex in CTRL and STING KO J774 cells through immunofluorescence. Upon OVA treatment, control J774 cells showed an accumulation of dots corresponding to MHC-I-OVA complexes, whereas in STING KO, this pattern was not found (Figure 3A,B), suggesting an impairment of SIINFEKL peptide loading onto MHC-I.

Because we found a strong reduction in both membrane (Figure 1A–C) and intracellular levels (Figure 3A,B) of the MHC-I-SIINFEKL complex in STING KO J774 compared to control cells, which could likely be explained by an impairment of SIINFEKL peptide loading onto MHC-I, we asked whether the absence of STING in J774 cells could affect the expression of the main proteins of the peptide-loading complex, including TAP1, TAP2, and TPN. To this end, we evaluated the protein expression of TAP1, TAP2, and TPN in STING KO and control J774 cells after OVA stimulation [20] for up to 24 h. We did not observe any induction in the expression of TAP1, TAP2, or TPN proteins upon OVA treatment in control J774 cells (Figure 4A,B) while, interestingly, we observed a statistically significant increase in basal expression as well as an induction of TAP1 and TAP2 proteins upon OVA treatment in STING KO cells (Figure 4A,B), likely due to a compensatory mechanism. Of note, although STING KO J774 cells showed an increase in TAP1 and TAP2 protein contents both in basal condition and upon OVA treatment, they also showed a strong reduction in both membrane (Figure 1A–C) and intracellular levels (Figure 3A,B) of the MHC-I-SIINFEKL complex.

### 2.3. STING Is Required for an Efficient Expression of β2-Microglobulin and for the Formation of the MHC-I Complex

Next, we hypothesized that the absence of STING could affect the total amount of MHC-I as well as of its components, including the H2-K1 heavy chain (the principal heavy chain involved in OVA presentation) [21] and the β2-microglobulin (β2 m) light chain. To address this point, we evaluated the amount of MHC-I on the membrane of STING KO or control J774 cells, in the presence or absence of OVA treatment (500 µg/mL) for 24 h. After cell staining with anti-MHC-I-FITC or IgG-FITC as a control, we found a strong reduction in median fluorescence intensity (MFI) of MHC-I (Figure 5A) as well as in the percentage of MHC-I^+^ cells in both untreated and OVA-treated STING KO compared to control J774 cells (Figure 5B). In particular, J774 control cells showed about 80% MHC-I-positive cells at steady state, increasing to 98.5% after 24 h of treatment with OVA (Figure 5B). In contrast, the basal level of MHC-I on the cell surface of STING KO cells was deeply compromised (8.7%), with a slight increase after 24 h of treatment with OVA (30.5%) (Figure 5B). Based on these results, we further analyzed the mRNA expression of the main components of the MHC-I complex (H2-K1 and β2-microglobulin) at the steady state and upon OVA treatment in control and STING KO J774 cells. To this end, we measured the mRNA expression of H2-K1 and β2-microglobulin in untreated control and STING KO J774 cells or after OVA treatment (500 µg/mL). We did not found any statistical differences in H2-K1 mRNA expression at both steady state level and upon OVA treatment in control and STING KO J774 cells (Figure 5C), while we found an above three-fold reduction in β2-microglobulin mRNA expression in STING KO cells compared to control J774 cells at all the experimental points (Figure 5D). In particular, we observed that in control J774 cells, OVA treatment induced β2-microglobulin mRNA expression with a peak at 4 h, which gradually decreased at basal level 24 h post treatment, while in STING KO J774 cells, we did not observe any statistically significant induction in β2-microglobulin mRNA expression by OVA treatment (Figure 5D). Next, we evaluated the protein content of β2-microglobulin in control and STING KO J774 cells treated with or without OVA. Consistent with the results on β2-microglobulin mRNA expression, we observed a strong decrease in the amount of β2-microglobulin protein STING KO compared to J774 control cells in the presence or absence of OVA treatment; however, we did not observe any statistically significant induction of β2-microglobulin protein expression upon OVA treatment in either control or STING KO J774 cells (Figure 5E,F). Taken together, these data suggested that STING is essential for the expression of β2-microglobulin, a major component of MHC-I.

### 2.4. The Lack of STING Impairs the JAK/STAT Signaling

Next, we focused on the molecular mechanism through which STING could regulate the expression of β2-microglobulin and consequently of MHC-I. The signaling induced by JAK/STATs is critical for MHC-I expression [22]. It is worth noting that the β2 m promoter harbors several ISRE (interferon-sensitive response element) enhancers [23,24], which are binding sites for JAK activated STAT1 dimers [22] and are essential for β2 m gene transcription and expression. Therefore, we attempted to dissect JAK/STAT signaling in the presence or absence of STING. To this end, we treated control and STING KO J774 cells with or without OVA (500 µg/mL) for the indicated time points and consequently analyzed the phosphorylation status of JAK1 (Y1034/Y1035) and STAT1 (Y701) proteins as well as their total content (Figure 6A, Appendix A). We found that in control J774 cells, OVA treatment induced JAK1 and STAT1 phosphorylation and, in particular, the kinetics of STAT1 phosphorylation showed a strong peak at 4 h that decreased at 24 h, while that of JAK1 phosphorylation showed an increase at 4 h, which remained constant up to 24 h (Figure 6A, Appendix A). In addition, an induction of STAT1 total protein content was also observed at 24 h (Figure 6A, Appendix A). Interestingly, in STING KO J774 cells, the induction of both STAT1 and JAK1 phosphorylation by OVA treatment was impaired and, in addition, we observed a three-fold reduction in STAT1 total content and a two-fold reduction of JAK1 (Figure 6A, Appendix A). Following the observation about the reduction in STAT1 total content in STING KO J774 cells, we wondered whether the absence of STING could affect the mRNA expression of STAT1. To address this question, we analyzed the mRNA expression of STAT1 in control and STING KO J774 cells in the presence or absence of OVA treatment (500 µg/mL) (Figure 6B). We observed a strong reduction in STAT1 mRNA expression in STING KO compared to control J774 cells at all the experimental points. Moreover, we found that in the presence of STING, OVA treatment induced STAT1 mRNA expression, with a peak at 4 h, which gradually decreased up to 24 h, while in absence of STING, we did not observe any induction of STAT1 mRNA expression (Figure 6B).

Because STAT1 activation represents one of the most important hallmarks of innate immunity, we asked whether the molecular mechanism of STING-dependent STAT1 activation also occurred upon LPS stimulation. To this end, we treated control and STING KO J774 cells with or without LPS (1 µg/mL) for the indicated time points and analyzed the status of phosphorylation of JAK1 and STAT1 as well as their total content (Figure 6C, Appendix A). Consistent with OVA treatment, LPS induced the phosphorylation of JAK1 and STAT1 in control J774 cells, while LPS failed to induce it in STING KO J774 cells, which, in addition, showed a minor content of total STAT1 (Figure 6C, Appendix A). Taken together, these results indicate that STING is essential for an efficient activation of the JAK/STAT pathway (Figure 6A–C, Appendix A) and for STAT1 transcriptional regulation (Figure 6B).

Next, we asked whether the hyper-expression of STAT1 could rescue the phenotype of STING KO J774 cells. To this end, we generated a plasmid for the expression of the fusion protein STAT1-GFP and transfected it or the empty vector plasmid into STING KO J774 cells through nucleofection (Appendix A). Next, we treated control and STING KO J774 cells, transfected with empty vector or STAT1-GFP plasmids, with or without OVA (500 µg/mL). Consistent with previous results (Figure 6A), in the absence of STING a strong reduction in STAT1 expression occurred and the STAT1 phosphorylation was abolished (Appendix A) and, in this condition, the over-expression of STAT1 was not able to rescue the phosphorylation levels of STAT1 (Appendix A). In addition, we observed a strong reduction in MHC-I and in MHC-I-SIINFEKL complex in the absence of STING upon OVA treatment (Appendix A), and the hyper-expression of STAT1 was unable to increase the content levels of MHC-I and of MHC-I-SIINFEKL complexes (Appendix A). In agreement with these results, Fludarabine treatment of J774 cells, a well-known inhibitor of STAT1 phosphorylation, reduced the phosphorylation status of STAT1 (Appendix A) as well as the amount of MHC-I-SIINFEKL complex in the presence or absence of OVA treatment (Appendix A). Taken together, these data suggest that the lack of STING affects STAT1 expression and its phosphorylation status at tyrosine Y701 and that the STAT1 phosphorylation is essential for the generation of MHC-I-SIINFEKL complex on the plasma membrane, required for an efficient antigen presentation (Figure 6, Appendix A).

## 3. Discussion

Because STING has a central role in the regulation of immunity upon infection [25] and has the capability to trigger anti-tumor effects in response to neoantigens and chemo-radiotherapy [26], we decided to investigate its role in the process of MHC-I restricted antigen presentation.

In order to counteract infection by foreign pathogens, cells respond to the infection by processing exogenous hosts and exposing them to the plasma membrane together with MHC molecules. Depending on the origin of the antigens, the process can occur via MHC class I or class II. Cytoplasmatic antigens, derived by host infection, are processed via proteasome and transferred to the ER to be loaded onto MHC-I. The expression on the plasma membrane of the complex generates a “kill me” signal that triggers CD8^+^ T-cell cytotoxic action against the infected cell [7].

Of note, exogenous antigens, derived from the phagocytosis of infected cells, generate an MHC-II-dependent antigen presentation. The MHC-II-Ag complex on the membrane activates a CD4^+^-dependent humoral response. However, during the endocytic pathway, a small number of proteins can escape the vesicles and enter the cytoplasm. In this case, the antigens associate with MHC-I in a mechanism named cross presentation [7].

The experimental model we used was designed to detect MHC-I restricted antigen presentation. By using ovalbumin, we exploited the ability of J774 macrophages to internalize an exogenous protein, mimicking an infection from an exogenous pathogen. Upon entry, OVA is proteolyzed in order to generate the correspondent antigens to interact with MHC-I [14,15,16,17,18]. In particular, we analyzed the production of SIINFEKL by using an antibody raised against the MHC-I-SIINFEKL complex. We observed that the lack of STING impairs the MHC-I-dependent antigen presentation (Figure 1A–C). Of note, in agreement with our results, recently, Barnowski et al. found that bone marrow dendritic cells (BMDC) derived from both WT and STING KO mice incubated with feeder cells infected with an OVA-Modified Vaccinia Ankara virus (MVA) failed to present the MHC-I-SIINFEKL complex on the plasma membrane [27]. Next, we asked whether a defective uptake and proteolysis of OVA could occur in J774 STING KO cells; we assessed this question by using FITC-OVA and DQ-OVA to monitor the uptake and proteolysis of OVA, respectively (Figure 2A–D). We observed that the lack of STING did not affect the entry and proteolysis of OVA exogenous protein (Figure 2A–D), even though there appeared to be a slight decrease in the internalization of OVA, which could suggest a slight impairment of endocytosis due to the lack of STING (Figure 2B). It is worth mentioning that the activation of STING is related to the transcriptional induction of genes involved in several biological processes including response to stress, endocytosis, microtubule-based movement, mitochondrial fission, and carbohydrate metabolism [28]. Because we found a strong reduction in both membrane (Figure 2A,D) and intracellular levels (Figure 3A,B) of the MHC-I-SIINFEKL complex in STING KO J774 cells compared to control cells, we asked whether the absence of STING could affect the expression of the main proteins of the peptide-loading complex, including TAP1, TAP2, and TPN. Interestingly, we observed a statistically significant increase in the basal expression of TAP1 and TAP2 proteins in STING KO compared to control J774 cells (Figure 4A,B), likely due to a compensatory mechanism, even though the amount of MHC-I-SIINFEKL complex on both the membrane (Figure 2A–D) and intracellular levels (Figure 3A,B) were reduced.

Because the lack of STING does not affect either the entry/proteolysis of OVA nor the expression of the peptide-loading complex, we hypothesized that the absence of STING could affect the total amount of MHC-I as well as of its components, including the H2-K1 heavy chain (the principal heavy chain involved in OVA presentation) [29] and the β2-microglobulin (β2 m) light chain. Through the analysis of mRNA expression of H2-K1 and β2-microglobulin in control and STING KO J774 cells, in the presence or absence of OVA treatment, we found that the absence of STING caused a strong reduction in β2-microglobulin mRNA and protein expression (Figure 5D–F) (the light chain of MHC-I), but not in H2-K1 (Figure 5C). It is worth mentioning that in a recent paper focusing on the analysis of transcriptome modulation upon STING depletion, they observed a reduced expression in β2-microglobulin in STING -/- LLC and CT26 cell lines [25].

Considering that β2-microglobulin is necessary for the association of MHC-I heavy chain with the peptide-loading complex and for the loading of the antigens [30] and that in the absence of β2 m, MHC-I heavy chains are not able to relocate to the plasma membrane [31], the strong reduction in β2-microglobulin mRNA expression may likely explain the drastic reduction in MHC-I-SIINFEKL complex levels on both the membrane (Figure 2A–D) and in the cytoplasm (Figure 3A,B) as well as the reduction in the amount of MHC-I on the membrane (Figure 5A,B).

Drew et al. demonstrated that gene expression regulation of MHC-I components is triggered by NF-κB and IRF1 and the activity of the latter depends on the activation of JAK/STAT signaling [22]. Because of its central role in inducing the antigen presentation, we analyzed the JAK1/STAT1 activation upon OVA or LPS treatment. We found that in the absence of STING, the induction of both STAT1 (Y701) and JAK1 (Y1034/Y1035) phosphorylation by OVA and LPS treatment was impaired, and, in addition, we observed a three-fold reduction of STAT1 total content (Figure 6A–C, Appendix A). In agreement with these results, in the presence of STING, OVA treatment induced STAT1 mRNA expression, while in the absence of STING, the basal level of STAT1 mRNA expression was strongly reduced and OVA treatment did not cause any induction of STAT1 mRNA expression (Figure 6B). Interestingly, we observed that in the absence of STING, the overexpression of STAT1, which was not accompanied by an increase in its phosphorylation status, was not able to rescue the MHCI levels and the antigen presentation rate, suggesting that the STING-dependent phosphorylation of STAT1 is essential for an efficient MHC class I antigen presentation (Appendix A). Consistent with these findings, we observed that treatment with Fludarabine, a well-known inhibitor of STAT1 phosphorylation [32], negatively affected the MHC class I antigen presentation of OVA257–264 (Appendix A).

The recent literature has highlighted a central role of STING in regulating the anti-tumor immune response in a microbiota-dependent manner. In particular, Lam KC et al. showed that microbiota activate the anti-tumor immune response of mononuclear phagocytes, including monocytes, macrophages, and dendritic cells in the tumor microenvironment, through the activation of STING [33]. Consistent with this report, Li Z., et al. showed that gut microbiome dysbiosis impairs antitumor immune responses by suppressing antigen presentation and inhibiting effector T-cell effector functions through the cGAS-STING pathway [34]. Our data provide a direct link between the absence of STING and the impairment of MHC-I antigen presentation via the inhibition of β2-microglobulin gene expression and STAT-1 activation.

In conclusion, our analysis reveals a central role of STING in regulating antigen presentation and provides a new piece in the extremely intricate puzzle of immune response. Our findings could be extremely relevant in the pathogenesis of several diseases and potentially provide novel therapeutic targets and strategies for autoimmune disorders or cancer.

## 4. Materials and Methods

### 4.1. Chemicals

Chemicals were purchased from the following manufacturers: Dulbecco’s modified Eagle’s medium (DMEM), fetal bovine serum (FBS), penicillin-streptomycin, phosphate buffer saline (PBS), and Trypsin-EDTA 0.25% from GIBCO (Thermo Fisher Scientific, Inc., Waltham, MA, USA); protease and phosphatase inhibitors cocktails from Roche Diagnostic (Meylan, France); protein assay and acrylamide/bis-acrylamide solution 40% from Bio-rad (Munchen, Germany); ECL from Elabscience (Houston, TX, USA); PagerulerTM Protein Ladder, Alexa FluorTM 488-conjugated ovalbumin, DQ-ovalbumin, and ProLongTM Gold Antifade Mountant with DAPI from Invitrogen-Thermo Fisher Scientific; and formaldehyde solution 37%, glycine, ovalbumin (OVA), lipopolysaccharides (LPS), and Fludarabine from Sigma-Aldrich (St. Luis, MI, USA).

### 4.2. Antibodies

The following antibodies were purchased: rabbit polyclonal anti STING #19851-a-AP (Proteintech, Rosemont, IL, USA), mouse polyclonal anti-STAT1 A302-753A (Bethyl Laboratories Inc., Montgomery, TX, USA), mouse monoclonal anti p-STAT1 (Y701) sc-136229, mouse monoclonal anti TAP1 (D-11) sc-518133, mouse monoclonal anti TAP2 (B-2) sc-515576, mouse monoclonal anti TPN (TO-3) (Santa Cruz Biotechnology, Dallas, TX, USA), rabbit monoclonal anti beta-2 microglobulin (4H5L6) (Thermo Fisher Scientific) #701250, mouse monoclonal anti γ-tubulin #T6657 (Sigma-Aldrich, St. Louis, MI, USA), FITC mouse monoclonal MHC Class I (H-2Kb) (AF6-88.5.5.3) #11-5958-83, PE mouse monoclonal MHC Class I (H-2Kb) (AF6-88.5.5.3) # 12-5958-82, PE mouse monoclonal SIINFEKL/H-2Kb (eBio25-D1.16) #12-5743-82, PE mouse monoclonal IgG1 kappa isotype control (P3.6.2.8.1) #12-4714-42, FITC mouse monoclonal IgG2a kappa (eBM2a) #11-4724-81 (eBioscience^TM^, Thermo Fisher Scientific), and horseradish peroxidase (HRP)-conjugated secondary antibodies GtxRb-003-DHRPX and GtxMu-003-DHRPX (Immuno Reagents Inc., Raleigh, NC, USA).

### 4.3. Plasmids

The plasmids for knocking out STING were generated as follows: Guide RNAs were cloned in lentiCRISP v2 (addgene #52961) following BsmBI digestion. The following primers were used to generate the gRNAs: STING gRNA1 forward, 5′-CACCGCACCTAGCCTCGCACGAACT-3′; STING gRNA1 reverse, 5′-AAACAGTTCGTGCGAGGCTAGGTGC-3′; STING gRNA2 forward, 5′-CACCGATATTTGGAGCGGTGACCTC-3′; STING gRNA2 reverse, 5′-AAACGAGGTCACCGCTCCAAATATC-3′. Double stranded DNA fragments were obtained by preparing a mixture containing 10 μM primer forward, 10 μM primer reverse, 50 mM Tris-HCl, 10 mM MgCl_2_, 10 mM Dithiothreitol, and 1 mM ATP. The mixture was incubated for 30 min at 37 °C followed by 5 min at 95 °C and then slowly cooled down to room temperature. Double-stranded gRNAs were digested with BsmBI and cloned into lentiCRISP v2. To generate pLenti-C-STAT1-mGFP-P2A-Puro. The STAT1 coding sequence was amplified with RT-PCR from J774 cDNA and ligated to BamHI/XhoI digested pLenti-C-mGFP-P2A-Puro. The following primers were used for the amplification: STAT1 BamHI forward, 5′-ACTGGGATCCTCAAGGATGTCACAGTGGTT-3′; STAT1 XhoI reverse, 5′-ACTGCTCGAGTACTGTGCTCATCATACTGTCA-3′.

### 4.4. Cell Culture, Treatments, and Viral Transductions

J774 and HEK293T were cultured in DMEM supplemented with 10% heat-inactivated FBS, 2 mM glutamine, and 0.5 mg/mL of penicillin-streptomycin. All the cultures were maintained in a humidified incubator at 37 °C with 5% CO_2_.

STING knock-out was carried out by CRISPR/Cas9 by using lentiviral particles derived from lentiCRISP v2 as follows.

Lentiviral particles expressing STING gRNAs or empty vector were produced by transfection of HEK293T cells. Briefly, the HEK293T cells (2 × 10^6^) were transfected with lentiCRISP v2 (10.5 μg), pMD2G (3.8 μg), pMDLg-RRE (7 μg), and pRSV-Rev (2.6 μg). The cell culture media were changed the day after transfection, and the supernatants were harvested 48 h post transfection and filtered through a 0.45 μm filter (Millipore, Billerica, MA, USA). J774 were infected either with lentiCRISP v2 CTRL or with a 1:1 ratio of the lentiCRISP v2 STING gRNAs or single gRNAs. Puromycin selection was used to isolate stably transduced cells. Knock-out was assessed by western blot analysis. STAT1 overexpression was carried out by using lentiviral particles as above. As a negative control, STING KO cells were infected with the empty vector.

Ovalbumin treatments were performed by incubating the cells with 500 μg/mL of OVA for the indicated time. As a negative control, cells were treated with the same volume of PBS. For LPS treatments, cells were incubated with 1 μg/mL for the indicated time. An equal volume amount of DMSO was added to the untreated cells as a negative control. Fludarabine experiments were performed as follows. Cells were pre-treated with 25 ng/mL of Fludarabine for 6 h following OVA treatments as above.

### 4.5. Western Blot Analysis

Cells were lysed in RIPA Buffer (150 mM NaCl, 2% NP40, 0.1% SDS, 50 mM Tris HCl) supplemented with protease and phosphatase inhibitors on ice for 30 min [35]. Upon centrifugation, protein concentrations were evaluated by Bio-rad assay. Proteins (30 μg) were resuspended in Laemmli buffer (4% SDS, 10% β-mercaptoethanol, 20% glycerol, 0.125 M Tris HCl, 0.004% bromphenol blue) and resolved by SDS-PAGE. Proteins were transferred on Immobilon^®^-NC Transfer Membrane (Merck-Millipore, Burlington, MA, USA) and blocking was performed in 5% non-fat milk (neoFroxx GmbH, Einhausen, Germany). After three washes with TBS, membranes were immunoblotted overnight with primary antibodies at 4 °C. Incubation with secondary antibody was performed for 1 h at room temperature. After three washes with TBS Tween20 (0.1%), enhanced chemiluminescence (ECL) was used for protein detection. Image-J software was used for the densitometric analysis.

### 4.6. Uptake and Proteolysis Assay

J774 CTRL and J774 STING KO were seeded at 5 × 10^4^ cells on 0.22 mm glass coverslip in 12 MW and incubated overnight in standard culture conditions. For the uptake assay, cells were washed three times with PBS 1× and then incubated for 30 min with 50 μg/mL of Alexa Fluor^TM^ 488-conjugated ovalbumin (Thermo Fisher Scientific) in serum-free DMEM at 37 °C. Negative controls were performed on ice as above. Cells were washed three times with PBS, fixed, and processed for immunofluorescence.

A proteolysis assay was performed in serum-free DMEM. Cells were washed three times with PBS and pulsed for 3 min with 50 μg/mL of DQ-ovalbumin (Thermo Fisher Scientific). After three washes with PBS cells, were incubated in serum-free medium for 30 min at 37 °C. Negative controls were carried out on ice as above. After incubation, cells were washed three times with PBS, fixed, and processed for immunofluorescence.

### 4.7. Fluorescence Microscopy

Cells were seeded at 5 × 10^4^ on glass coverslips and incubated overnight under standard growth conditions. The next day, cells were rinsed three times with PBS and fixed in 3.7% formaldehyde for 30 min at room temperature. Formaldehyde quenching was performed by 10 min of room temperature incubation with glycine 0.1 M. Cells were permeabilized by incubation with blocking buffer (1% BSA, 0.01% Sodium Azide and 0.02% saponin in PBS) for 10 min at room temperature [36]. Immunostaining was carried out at room temperature for 1 h. Following three washes with PBS, cells were incubated with secondary antibodies for 30 min at room temperature. Finally, coverslips were rinsed with distilled water and mounted onto glass slides with the Prolong Gold anti-fade reagent with DAPI (Invitrogen, Thermo Fisher Scientific). Images were collected by using a laser scanning microscope (LEICA DMi8) and analyzed using LEICA LAS X software.

### 4.8. Flow Cytometry

Cells were detached from the plate by using PBS-EDTA 0.5 M and labeled with primary antibody for 30 min at 4 °C in the dark. After three washes with PBS, cells were analyzed with BD AccuriTM C6 cytometer (BD Biosciences, San Jose, CA, USA) [37]. A relative Ig isotype antibody was used as a control for non-specific binding. Data were collected by using BD Accuri C6 software (BD Biosciences).

### 4.9. Quantitative Real-Time PCR (RT-qPCR)

Total RNA was extracted using TRIzol reagent (Invitrogen) as per manufacturer’s instructions. A 1 μg amount of RNA was used for cDNA synthesis with M-Mulv Reverse Transcriptase (New England Biolabs Ipswich, MA, USA). The reaction was carried out as follows: 25 °C for 5 min, 42 °C for 1 h, 65 °C for 20 min. Real-Time PCR was performed by using SensiFAST^TM^ Sybr Green (Bioline, Meridian Bioscience, Cincinnati, OH, USA) with QuantStudio^TM^ 7 (Applied Biosystem). Reactions were carried out in triplicate in at least three independent experiments. Target expressions were normalized against the housekeeper gene GAPDH and analyzed with the 2^−ΔΔCT^ method [38,39]. Fold changes in gene expression were normalized to an internal control (J774 CTRL NT). The primer-BLAST tool (https://www.ncbi.nlm.nih.gov/tools/primer-blast/, accessed on 28 July 2022) was used for the design of primers which were purchased from Eurofins Genomics. The following primers were used to amplify the target genes: STAT1 forward, 5′-GTTCCGACACCTGCAACTGAA-3′; STAT1 reverse, 5′-ACGACAGGAAGAGAGGTGGT-3′; H2K1 forward, 5′-CAGGTGGAAAAGGAGGGGAC-3′; H2K1 reverse, 5′-CTGAGGGCTCTGGATGTCAC-3′; β2 m forward, 5′-GACCGGCCTGTATGCTATCC-3′; β2 m reverse, 5′-TGTCTCGATCCCAGTAGACG-3′; GAPDH forward, 5′-GTATGACTCCACTCACGGCAAA-3′; GAPDH reverse, 5′-TTCCCATTCTCGGCCTTG-3′.

### 4.10. Statistical Analysis

Our data analysis covers at least three independent experiments. Statistics were performed by Student’s *t*-test with significant *p*-value threshold <0.01.

## Figures and Tables

**Figure 1 ijms-23-14232-f001:**
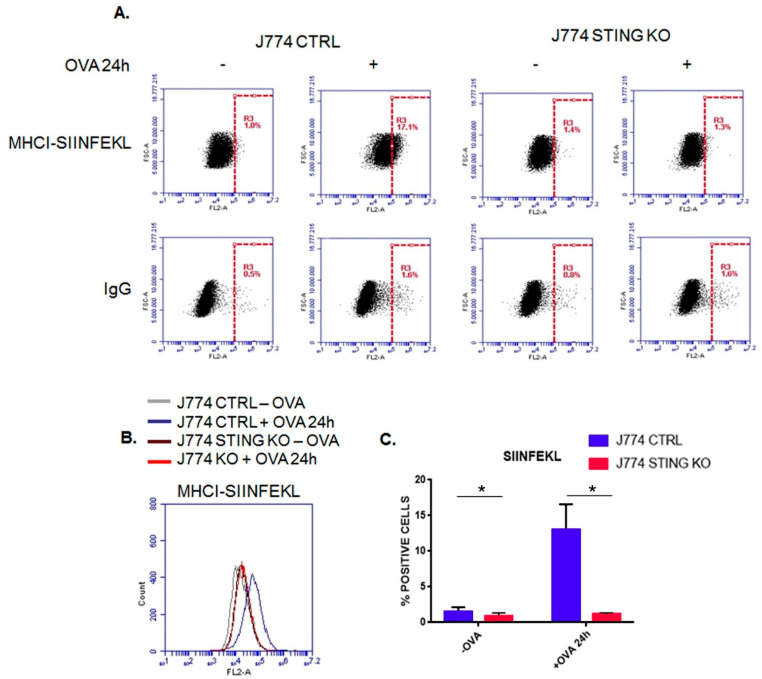
MHC-I-dependent antigen presentation of ovalbumin is impaired in STING KO macrophages. (**A**) J774 CTRL and STING KO (1 × 10^6^) were treated with 500 μg/mL of OVA for 24 h and were stained with SIINFEKL/H-2Kb-PE and IgG-PE as a control. Each plot represents 10,000 events of a representative experiment. (**B**) Histogram of J774 CTRL and STING KO treated as above. (**C**) Percentage of SIINFEKL/H-2Kb-positive population in untreated or OVA treated cells. Values (mean ± SE, n = 5) are shown. The asterisks indicate a statistically significant difference compared to the untreated control, according to a Student’s *t*-test (*p* < 0.01).

**Figure 2 ijms-23-14232-f002:**
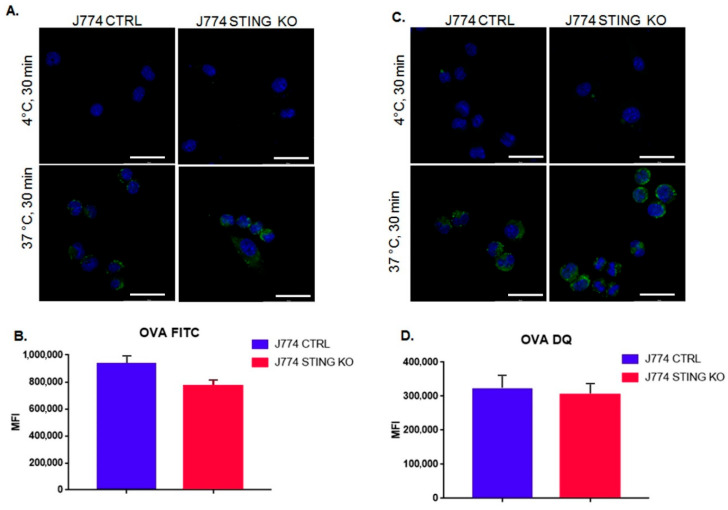
STING KO does not affect uptake and proteolysis of ovalbumin in murine macrophages. (**A**) J774 CTRL and J774 STING KO cells (2 × 10^5^) were incubated in serum-free medium with 50 μg/mL of Alexa 488-conjugated ovalbumin for 30 min at 37 °C or on ice as a negative control. Nuclei were stained with DAPI. Original magnification 20x and white scale bars represent the length of 5 μm. (**B**) Mean fluorescence intensity (MFI) of Alexa 488-conjugated OVA was analyzed by cytofluorimeter. Values (mean ± SE, n = 3) are shown. (**C**) J774 CTRL and STING KO (2 × 10^5^) were pulsed in serum-free medium with 50 μg/mL of DQ-ovalbumin for 3 min and chased for 30 min at 37 °C. Continuous incubation on ice was performed as a negative control. Nuclei were stained with DAPI. The original magnification is 20× and white scale bars represent a length of 5 μm. (**D**) Mean fluorescence intensity (MFI) of DQ-OVA signal was measured with a cytofluorimeter. Values (mean ± SE, n = 3) are shown.

**Figure 3 ijms-23-14232-f003:**
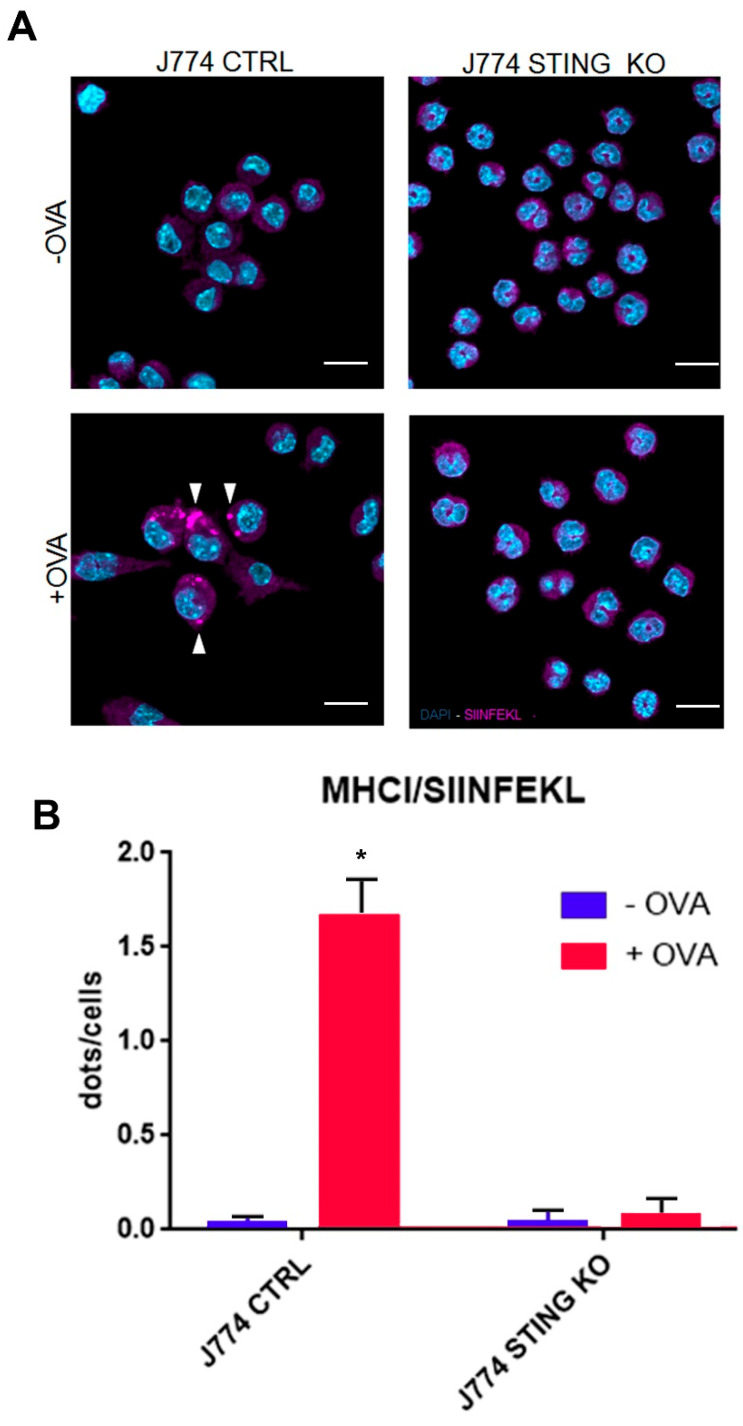
Peptide loading is impaired in STING KO macrophages. (**A**) J774 CTRL and STING KO cells (2 × 10^5^) were treated with 500 μg/mL of OVA for 24 h and stained with anti-MHC-I/SIINFEKL. Nuclei were stained with DAPI. Arrows indicate MHCI/SIINFEKL complexes. The original magnification is 20× and white scale bars represent a length of 5 μm. (**B**) A quantification analysis of MHCI/SIINFEKL dots was carried out by using a sample of 150 cells from three independent experiments. The asterisks indicate a statistically significant difference compared to the untreated control, according to a Student’s *t*-test (*p* < 0.01).

**Figure 4 ijms-23-14232-f004:**
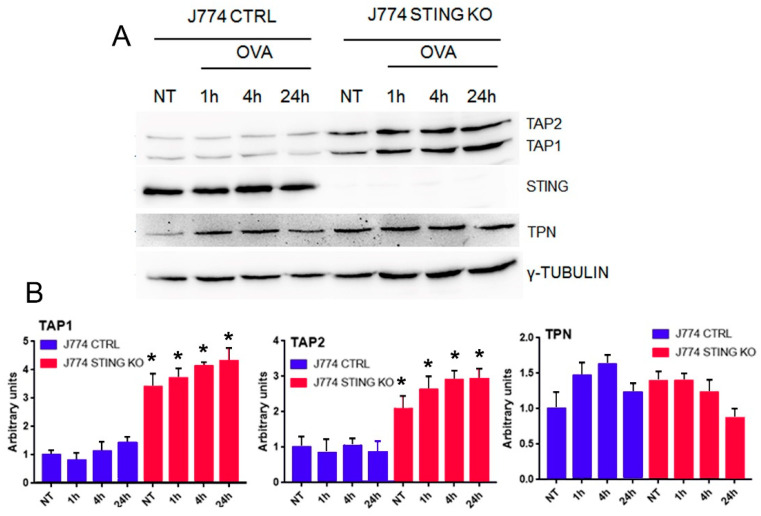
Protein expression of the components of the peptide-loading complex (TAP1, TAP2 and TPN) in control and STING KO macrophages. (**A**) J774 CTRL and STING KO cells (5 × 10^6^) were treated with 500 μg/mL of OVA or left untreated for the indicated time. Whole cell extracts (30 μg) were analyzed by western blot using the indicated antibodies. γ-Tubulin was included as a control for protein loading. (**B**) Quantifications of the protein levels of TAP1, TAP2 and TPN in CTRL (blue bars) and STING KO (red bars) were calculated by ImageJ software as fold-change relative to unstimulated, which was set to 1.0. Values (mean ± SE, n = 4) are shown. The asterisks indicate a statistically significant difference compared to the untreated control, according to a Student’s *t*-test (*p* < 0.01).

**Figure 5 ijms-23-14232-f005:**
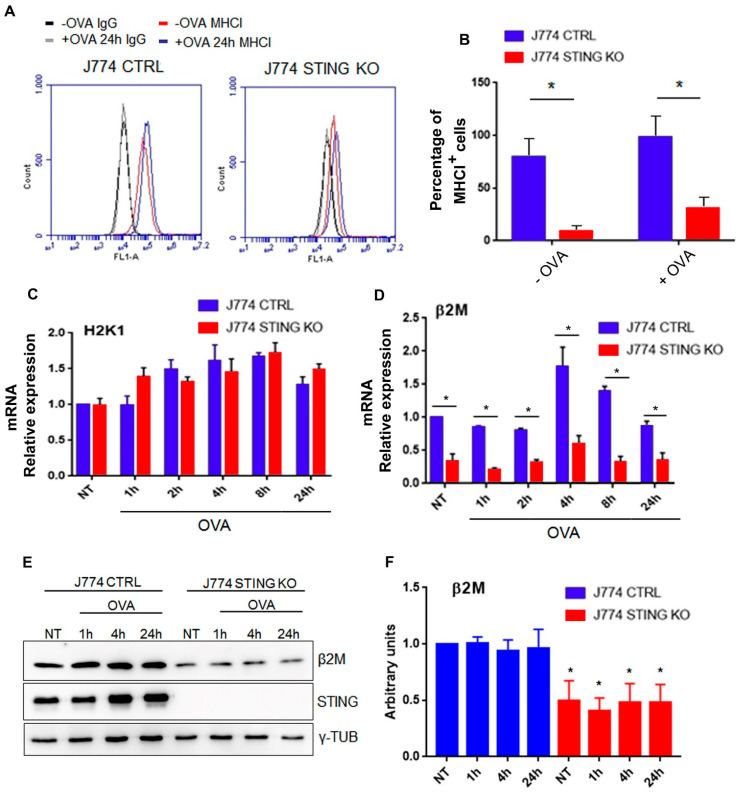
MHC-I expression on the plasma membrane and mRNA expression of H2K1 and β2 m in control and STING KO macrophages. (**A**) J774 CTRL and STING KO cells (1 × 10^6^) were treated with 500 μg/mL of OVA for 24 h and were stained with MHC-I-FITC and IgG-FITC as a control. Each plot represents 10,000 events of a representative experiment. (**B**) Percentage of MHC-I-FITC-positive population in untreated or OVA treated cells. Values (mean ± SE, n = 5) are shown. The asterisks indicate a statistically significant difference compared to the untreated control, according to a Student’s *t*-test (*p* < 0.01). (**C**,**D**) Cells (5 × 10^6^) were stimulated with 500 μg/mL of OVA for 1, 2, 4, 8, 24 h or left untreated. Total RNA was extracted and analyzed by RT-qPCR to evaluate the gene expression of H2K1 (**C**) and β2 m (**D**). Values are presented as the mean ± SD of three independent experiments. The asterisks indicate statistically significant differences between CTRL and KO cells, according to a Student’s t test (*p* < 0.01). (**E**) J774 CTRL and STING KO cells (5 × 10^6^) were treated with 500 μg/mL of OVA or left untreated for the indicated time. Whole cell extracts (30 μg) were analyzed by western blot using the indicated antibodies. γ-Tubulin was included as control for protein loading. (**F**) Quantifications of the protein levels of β2 m in CTRL (blue bars) and STING KO (red bars) cells were calculated by ImageJ software as fold-change relative to unstimulated, which was set to 1.0. Values (mean ± SE, n = 4) are shown.

**Figure 6 ijms-23-14232-f006:**
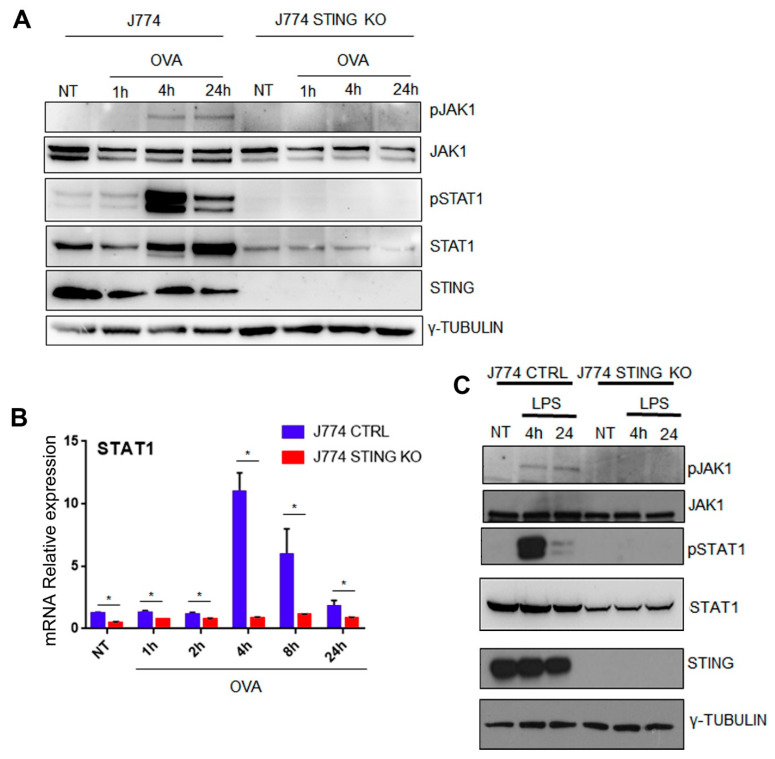
JAK1/STAT1 activation and STAT1 expression are impaired in STING KO macrophages upon OVA or LPS treatments. (**A**) J774 CTRL and STING KO cells (5 × 10^6^) were treated with 500 μg/mL of OVA or left untreated for the indicated time. Aliquots of whole cell extracts (30 μg) were analyzed by western blot for the indicated proteins. γ-Tubulin was included as a control of protein loading. (**B**) Cells (5 × 10^6^) were stimulated with 500 μg/mL of OVA for 1, 2, 4, 8, 24 h or left untreated. Total RNA was extracted and analyzed by RT-qPCR to evaluate the gene expression of STAT1. Values (mean ± SD = 3) are indicated. The asterisks indicate statistically significant differences between CTRL and KO cells, according to a Student’s t test (*p*  <  0.01). (**C**) Murine macrophage CTRL and STING KO were stimulated with 1 μg/mL of LPS for 4 and 24 h or DMSO as a negative control. Cells were lysed and whole protein extracts were blotted and analyzed with the indicated antibodies. Blots are representative of three independent experiments.

## Data Availability

The data presented in this study are available on request from the corresponding authors.

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
