# Peer review of "The Lack of STING Impairs the MHC-I Dependent Antigen Presentation and JAK/STAT Signaling in Murine Macrophages"

_ijms, 2022, doi:10.3390/ijms232214232_

Round 1
Reviewer 1 Report
This manuscript reported the function of macrophaage STING in MHC-I dependent antigen presenation via JAK/STAT signaling pathway. My major concerns are:
1. According to "Materials and Methods", two gRNAs for STING were used to knockout/knockdown macrophage STING expression via CRISPR/Cas9 system. It is not clear which one was used for which experimen(s). Indeed, to prevent off-target effects, all of the experiment shown in this manuscript, both gRNAs shall be used individually. In additon, data to shown the knockout/knockdown efficiency of gRNAs mentioned above shall be shown as Figure 1. Trace amount of STING was detected on the Western blotting of Figure 4A.
2. Lack of justification to do the Co-IP for TAP1 and STING as shown in Figure 3C. Either provide the rationale or delete this figure. Is it necessary to check the binding of TAP2 and STING by Co-IP?
3. Does overexpression of STAT1 can rescue the phenotypes of J774 STING KO cells?
Reviewer 2 Report
In this paper, Caiazza et al described a new role of STING in MHC-I dependent antigen presentation in murine macrophages. To do so, they analyzed the effect of STING depletion, done by CRISPR-Cas9, on OVA-SINFEKL MHC-l antigen presentation in the murine macrophage cell line J774. They found that STING depletion led to impaired antigen presentation, as well as a decreased level of MHC-I complex in the plasma membrane and that this was due to an impaired antigen loading process, likely associated with a decrease in β2m mRNA expression levels. Finally, to elucidate possible molecular mechanisms underlying these results they assessed JAK-STAT1 signaling and found no activation of STAT1 and JAK1 upon OVA and LPS treatment in STING KO macrophages, as well as a significant decrease in total STAT1, suggesting dampened activation of the immune response. With this article, the authors revealed a possible new molecular role of STING in immune mechanisms that can be further elucidated and applied in the pathogenesis of immune disorders. Overall, the manuscript is clear, easy to follow, and well-structured. The hypothesis and the aims are well-explained and clear, and the experimental design is appropriate to test it. Regarding the methodology, it’s detailed enough to allow the reproducibility of the results. Concerning the content itself, in my opinion, this study is not sufficient to draw such conclusions. I think that lacks some validation to strengthen their conclusions, such as an in vivo validation using for example a mice model in which STING is KO in macrophages, and more importantly the translation of these results to human models. Regarding the discussion I think it is missing some discussion about the results, it’s very repetitive from the results section, and also about what is known about the matter in other models.
Specific comments
1. Line 64: “TAPN” instead of TAP
2. Figure 1: did the authors check MHC-II-SIINFEKL in their models? In the control condition (PBS) how MHC-I-SIINFEKL is detected?
3. Figure 2: why 4˚C as a negative control?
4. Figure 2B: even though there are no statistically significant differences in the uptake between CTRL and KO cells, there is a clear tendency to show a lower uptake by KO macrophages, which should be mentioned.
5. Line 147: The authors mention here, and along the text, that they found a strong reduction of membrane levels of MHC-I-SIINFEKL complex in STING KO compared to control cells in Figure 2A-D, but in this figure, they only mention that uptake and proteolysis were evaluated.
6. Figure 4: the first part of the legend should be in bold italic.
7. Figure 4A-B: in lines 153-155 they say that they do not observe an induction in the expression of TAP1, TAP2, and TPN upon OVA treatment, however, this is not true for STING KO macrophages, in which if we compare the NT and 24h conditions there is a clear increase in these proteins.
8. Figure 4C: these results are from which cells? They say that use both CTRL and STING KO cells, but there is no point in using STING KO cells to evaluate STING interaction with TAP1.
9. Figure 5A: did the authors check total levels of MHC-I and not only at the membrane?
10. Line 206 to 208, and Figure 5C-D: The authors conclude that STING is essential for the expression of B2-microglobulin, but what about the protein levels? I think it is important to check that.
11. Figure 6: Since in Figure 2 the authors present the protein quantification levels, I think they should also have it here or in the supplementary results.
12. Figure 6A: in the legend, they say that used 3x106 cells, while in the legend of figure 4A they used 5x106 cells. Why such a difference for similar experiments?
13. Line 286 to 289: I think the author should have a reference for the model used.
Round 2
Reviewer 1 Report
Authors addressed my concerns.